# Rapid progression is associated with lymphoid follicle dysfunction in SIV-infected infant rhesus macaques

Matthew P. Wood[1], Chloe I. Jones [1], Adriana Lippy[1], Brian G. Oliver[1], Brynn Walund [1], Katherine A. Fancher [1], Bridget S. Fisher[1], Piper J. Wright[1], James T. Fuller[2], Patience Murapa [2,3], Jakob Habib [4,5], Maud Mavigner[4,5,6], Ann Chahroudi [4,5,6], D. Noah Sather [1], Deborah H. Fuller [2,3], Donald L. Sodora [1] *

1 Center for Global Infectious Disease Research, Seattle Children's Research Institute, Seattle, Washington, United States of America, 2 University of Washington Department of Microbiology, Seattle, Washington, United States of America, 3 Washington National Primate Research Center, Seattle Washington, United States of America, 4 Yerkes National Primate Research Center, Emory University School of Medicine, Atlanta, Georgia, United States of America, 5 Department of Pediatrics, Emory University School of Medicine, Atlanta, Georgia, United States of America, 6 Center for Childhood Infections and Vaccines of Children's Healthcare of Atlanta and Emory University, Atlanta, Georgia United States of America

* donald.sodora@seattlechildrens.org

**Data Availability Statement:** All relevant data are within the manuscript and its Supporting Information files.

## Abstract

HIV-infected infants are at an increased risk of progressing rapidly to AIDS in the first weeks of life. Here, we evaluated immunological and virological parameters in 25 SIV-infected infant rhesus macaques to understand the factors influencing a rapid disease outcome. Infant macaques were infected with SIVmac251 and monitored for 10 to 17 weeks post-infection. SIV-infected infants were divided into either typical (TypP) or rapid (RP) progressor groups based on levels of plasma anti-SIV antibody and viral load, with RP infants having low SIV-specific antibodies and high viral loads. Following SIV infection, 11 out of 25 infant macaques exhibited an RP phenotype. Interestingly, TypP had lower levels of total CD4 T cells, similar reductions in CD4/CD8 ratios and elevated activation of CD8 T cells, as measured by the levels of HLA-DR, compared to RP. Differences between the two groups were identified in other immune cell populations, including a failure to expand activated memory (CD21-CD27+) B cells in peripheral blood in RP infant macaques, as well as reduced levels of germinal center (GC) B cells and T follicular helper (Tfh) cells in spleens (4- and 10-weeks post-SIV). Reduced B cell proliferation in splenic germinal GCs was associated with increased SIV+ cell density and follicular type 1 interferon (IFN)-induced immune activation. Further analyses determined that at 2-weeks post SIV infection TypP infants exhibited elevated levels of the GC-inducing chemokine CXCL13 in plasma, as well as significantly lower levels of viral envelope diversity compared to RP infants. Our findings provide evidence that early viral and immunologic events following SIV infection contributes to impairment of B cells, Tfh cells and germinal center formation, ultimately impeding the development of SIV-specific antibody responses in rapidly progressing infant macaques.

**Funding:** The data generated in this manuscript was supported by grants from: National Institute of Dental and Craniofacial Research DE023047 to DLS, DE026336 to DLS and DNS, National Institute of Allergy and Infectious Diseases AI133706 to AC, University of Washington Department of Global Health (US) A1750915 to MPW. The funders had no role in study design, data collection and analysis, decision to publish, or preparation of the manuscript.

**Competing interests:** The authors have declared that no competing interests exist.

## Author summary

Despite significant reductions in vertical HIV transmission, nearly 100,000 children succumb to AIDS-related illnesses each year. Indeed, infants face a disproportionately higher risk of progressing to AIDS, with roughly half of HIV+ infants exhibiting a rapid progression to AIDS-associated morbidity and mortality. Here, we evaluated immunological and virological parameters in 25 simian immunodeficiency virus (SIV)-infected infant rhesus macaques to assess the factors that influence a rapid disease outcome. Infant macaques were infected with simian immunodeficiency virus (SIV) and divided into either typical (TypP) or rapid (RP) progressor groups. RP infants exhibited low levels of plasma anti-SIV antibody and high viral loads. Following SIV infection, 11 out of 25 infant macaques exhibited an RP phenotype with some exhibiting AIDS-related symptoms. This study provides evidence that the low levels of anti-SIV antibodies are associated with impairments to both B and T cells in both blood and lymphoid tissues. These changes are associated with the prolonged expression of type 1 interferons which may be impeding development of a healthy humoral immune response in these rapidly progressing SIV-infected infant macaques. These findings have implications regarding potential therapeutic approaches to prevent rapid progression in HIV infected infants.

## Introduction

Despite significant reductions in vertical HIV transmission, nearly 100,000 children succumb to AIDS-related illnesses each year [1]. This can be in part attributed to a disproportionately higher risk of progressing to AIDS, with roughly half of infected infants exhibiting rapid disease progression within the first two years of life [2–7]. Despite this observation being first reported early in the epidemic [4,8], factors influencing why some infants exhibit a rapid HIV progression phenotype have yet to be fully understood.

Acute HIV infection is characterized by exponential viral replication in blood and tissues, depletion of mucosal CD4 T cells, production of type 1 interferons (IFNs) and increased IFN-induced gene transcription [9]. This is typically followed by an expansion of activated B cells in the blood and lymph nodes and production of anti-HIV antibodies targeting viral envelope proteins [10]. While the initial reduction of viral replication coincides with antibody production, antibodies produced during this period lack virus neutralizing activity, and viral control at the onset of chronic infection is often attributed to CD8 T cell responses [11,12]. However, in infants and children who progress rapidly to AIDS, plasma viral loads remain elevated well beyond acute infection, resulting in sustained type 1 IFN production [13], increased levels of inflammatory immune mediators [14,15] and a failure to develop humoral HIV responses [4].

Infection of macaques with SIV, in a minority of animals, results in rapid disease progression which is phenotypically consistent with rapid HIV infection in infant humans. This SIV-macaque rapid disease outcome, which includes high plasma viral loads and a lack of virus-specific plasma antibodies, has been documented in both adult [16–20], as well as infant macaques [21–24]. It is interesting that a sustained elevated plasma SIV viral load is associated with an unresolved type 1 interferon (IFN) response in blood and lymphoid organs [25,26]. These findings in SIV infected macaques contrasts with data obtained in nonpathogenic SIV natural host primate species such as African green monkeys (AGMs) and sooty mangabeys in which the type 1 IFN response is only transient, coinciding with the acute phase of the infection [20,27–29]. A study by Campillo-Gimenez et al. identified an increase in type I IFN production in both AGM and Controller Chinese rhesus macaques during the acute phase of the infection, which resolved to pre-infection baseline levels by 60 days post-infection [20]. In

contrast, Non-controller Chinese rhesus macaques, with an unresolved high viral load, maintained elevated type I IFN associated with increased levels of plasmacytoid dendritic cells (pDCs) in the lymph nodes [20]. Furthermore, rapid disease progression is associated with a failure of adult rhesus macaques to elicit SIV-specific antibody and induce B cell activation [18,19,30]. Indeed, SIV-specific antibody levels at 13 weeks post-infection strongly correlate with macaque survival [18]. These studies establish an association between prolonged elevated viral loads and type 1 IFN responses, low levels of SIV-specific antibody responses, and rapid disease progression.

Here we assess factors associated with viral load, SIV specific antibody responses and type 1 IFN responses during SIV-infection of infant rhesus macaques. We observe that of the twenty five SIV infected Indian origin infant macaques evaluated, eleven exhibit high levels of plasma SIV viral loads and low or undetectable levels of SIV specific antibodies, a phenotype consistent with rapid progression (RP). In contrast, 14 infant macaques exhibited a more typical disease progression phenotype (TypP), similar to that frequently observed in adult animals. Our findings support a model wherein early events following infection, including plasma viral load and genetic diversity of SIV, drive aberrant type 1 IFN responses, and that these are associated with lymphoid dysfunction and failure to develop an anti-SIV humoral immune responses in rapidly progressing SIV-infected infants.

## Results

### Oral SIV infection in macaque infants results in similar proportions of infants with rapid and typical disease progression

To evaluate the pathogenic outcome following oral mucosal SIV infection, infant rhesus macaques were infected through a series of escalating dose oral challenges with SIVmac251 and monitored for up to 22 weeks (median 10 weeks) (Fig 1A). SIV plasma viral load was measured throughout the time course (Fig 1B), and we focused on time points that reflect acute (2 weeks post-SIV), acute-chronic transition (4–6 weeks post-SIV) and the chronic phase (10+ weeks) of infection to assess differences over time. (Fig 1C). Plasma levels of anti-gp130 (Env)-specific IgG (Fig 1D) and IgA (Fig 1E) antibodies were evaluated at weeks 0, 4–6, and 10–12 post-SIV. Based on these observations, SIV-infected infant macaques can be divided into rapid progressors (RP) being those with the highest viral loads after week 2 and undetectable or very low levels of SIV-Env specific IgG and IgA antibodies during chronic infection. Factors that were not associated with rapid disease progression included SIV dose, sex or previous vaccination from a separate study (Table 1). Three of the fourteen TypP infants possessed the *Mamu B*008* MHC-1 haplotype, which has previously been associated with reduced SIV replication [31], however outcomes could not be explained entirely by infant MHC-1 allele. Age also did not appear to be a factor contributing to disease progression, with median ages of SIV acquisition for TypP and RP macaques being 11 weeks and 13 weeks, respectively. Additionally, 8 of 11 RP macaques developed clinical signs of simian AIDS prior to the end of the study follow-up, based on reports from veterinary staff. All infants showing clinical signs of AIDS exhibited either chronic SIV-associated diarrheal disease, failure to gain weight and wasting. This finding is consistent with earlier reports of SIV pathology in rapidly progressing macaques, with severe enteropathy and wasting observed in the absence of opportunistic infections [32].

### Acute stage levels of unique plasma viral variants negatively correlate with chronic stage anti-SIV antibodies

Based on previous work linking viral genetic diversity at acute infection with disease outcome [33], we evaluated viral diversity by the number of V1V2 Env variants during acute infection.

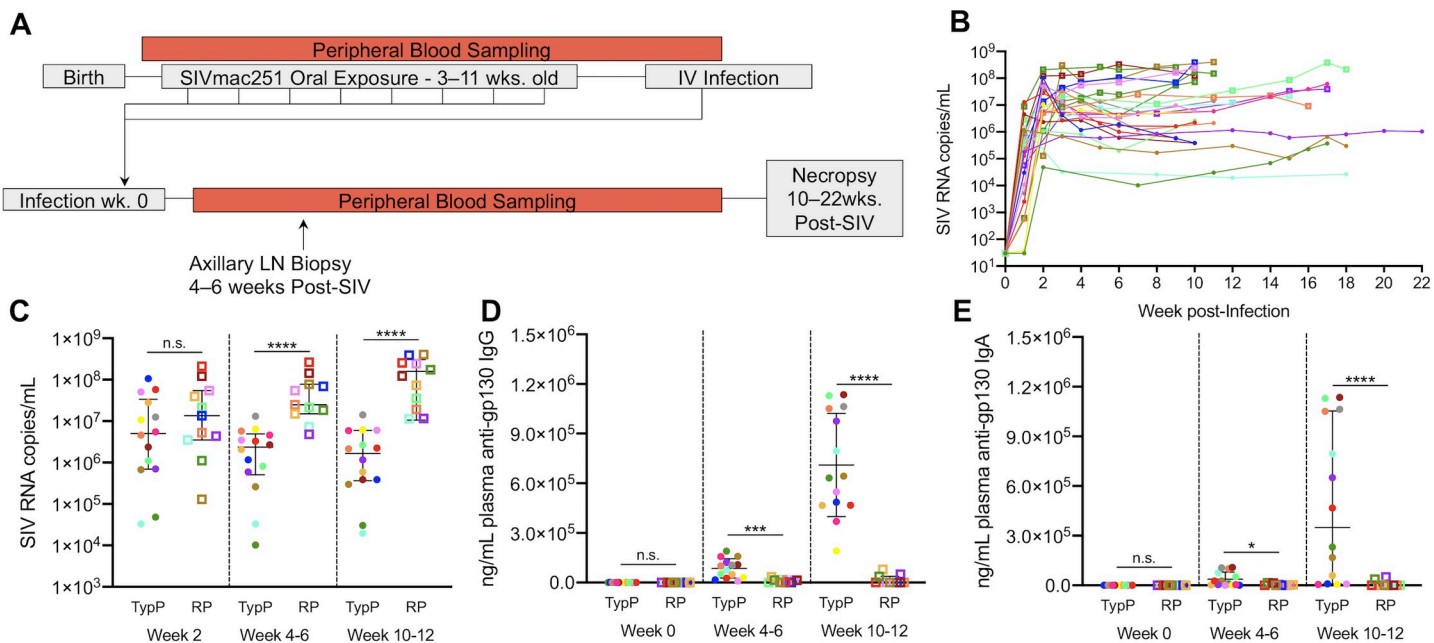

**Fig 1. SIV-infected infant macaques exhibit disparate anti-SIV IgG responses.** Infants were infected with SIVmac251 either orally (n = 22) or intravenously (n = 3) with SIVmac251 and viral load and production of anti gp120 antibodies was monitored until necropsy at 10–22 weeks post infection. 11 of 25 infants that did not produce anti-gp140 specific plasma IgG or IgA during chronic SIV infection (10–12 weeks post-SIV infection) and were characterized as Rapid Progressors (**RP**) (**A,B**). RP infants failed to reduce plasma viral load after acute infection and maintained higher viral loads during chronic infection (**D,E**). Typical progressors (**TypP**) are represented as **closed circles** and RP infants are represented as **open squares**. Statistical tests used to compare infant groups were carried out as described in the methods ** = p<0.01, *** = p<0.001, **** = p<0.0001. Error bars are shown as either mean with standard deviation or median with interquartile range based on data distribution.

Plasma was obtained at 2 weeks post-SIV infection and viral diversity within V1V2 region of SIV envelope assessed by Illumina MiSeq. This V1V2 478 bp region was selected as it contained the highest degree of DNA sequence variability within our SIVmac251 challenge strain. Representative phylogenies constructed by maximum likelihood are depicted for TypP (**Fig 2A**) and RP (**Fig 2B**) infants. After enumerating unique variant clusters in each infant group, we observed a significantly greater number of variants present at 2 weeks post-SIV in the RP compared to TypP macaques (**Fig 2C**). We also observed an inverse correlation between the anti-SIV-ENV IgG level (at week 10–12) and the number of observed V1V2 variant clusters (**Fig 2D**). Phylogenetic assessment of V1V2 variants in the SIVmac251 challenge stock and infant macaque plasma revealed that variants from RP infants are represented across a greater number of clades sharing sequences with the challenge stock than those from TypP infants (**S1A and S1B Fig**), indicating a more diverse genetic background in RP infant-derived plasma SIV. Greater genetic diversity in V1V2 of RP infant macaques suggests that RP infants are either infected with a greater number of SIV founders originating from the challenge stock, or that when macaques are initially infected, there are increased levels of SIV replication, and consequently accumulated mutations, within RP infants at 2 weeks post-SIV infection compared to the TypP infants.

## Total CD4 T cell levels are sustained during chronic infection of RP infants

To assess CD4 T cell depletion in SIV-infected infants, numbers of CD4 and CD8 T cells per mL of whole blood were used to calculate the CD4/CD8 ratio over time. Surprisingly, similar CD4/CD8 ratios were observed in both groups of infants at each time point examined (**Fig 3A and 3B**). Assessment of total CD4+ T cells (CD4+, CD3+) in peripheral blood revealed that

**Table 1. Study Animals.**

| Infant ID | Sex | MHC Class I Alleles | BCG Vaccinated | Disease Group | Age SIV+ (wks) | Dose SIV+ (TCID50) | Viral Load at Necropsy |
|---|---|---|---|---|---|---|---|
| A14112 | M | A1*006, A1*006, B*012, B*048 | Yes—Chronic SIV | TypP | 14 | 10000 | 3.70E+05 |
| A14113 | F | *A1*001*, A1*004, B*001, B*024 | Yes—Chronic SIV | TypP | 14 | 10000 | 6.17E+07 |
| A14206 | F | A1*004, A1*023, B*012, B*028 | Yes—Chronic SIV | TypP | 8 | 4000 | 3.01E+05 |
| A15068 | F | – | Yes—8 wks of age | TypP | 17 | 4000 | 3.94E+07 |
| A15187 | M | – | Yes—Chronic SIV | TypP | 11 | 4000 | 2.65E+04 |
| A16080 | F | A1*002, B*015 | No | TypP | 10 | 12000 | 6.03E+06 |
| A16083 | M | A1*026, A1*002, B*001, B*055 | No | TypP | 11 | 15000 | 3.88E+05 |
| A17151 | F | *A1*001*, A1*008, B*024, B*045 | No | TypP | 11 | 12000 | 2.14E+06 |
| A17153 | F | A1*023, A1*025, **B*008**, B*028 | No | TypP | 6 | 1000 | 3.85E+05 |
| A17154 | F | A1*002, A1*004, B*001 | No | TypP | 10 | 12000 | 1.41E+07 |
| A16185 | F | A1*008, A1*032, B*012, B*068 | Yes—1–2 wks of age | TypP | 10 | 12000 | 6.18E+06 |
| A16189 | M | A1*012, A1*025, **B*008** | Yes—1–2 wks of age | TypP | 15 | IV | 5.95E+05 |
| A17097 | F | A1*008, A1*011, B*001, **B*008** | Yes—1–2 wks of age | TypP | 10 | 12000 | 2.23E+06 |
| A17100 | M | *A1*001*, A1*023, B*002, B*028 | Yes—1–2 wks of age | TypP | 13 | 20000 | 2.63E+06 |
| A14114 | M | A1*004, A1*008, B*001, B*023 | Yes—Chronic SIV | RP | 18 | 4000 | 2.15E+08 |
| A14115 | M | A1*008, A1*012, B*012, B*055 | Yes—Chronic SIV | RP | 14 | 10000 | 9.18E+06 |
| A14203 | M | A1*002, A1*011, B*012, B*001 | Yes—Chronic SIV | RP | 8 | 4000 | 2.20E+07 |
| A15067 | F | – | Yes—8 wks of age | RP | 17 | 4000 | 1.04E+06 |
| A16079 | F | A1*004, A1*023, B*001, B*012 | No | RP | 12 | 20000 | 2.43E+08 |
| A16081 | F | A1*002, A1*006, B*012, B*024 | No | RP | 15 | IV | 3.91E+08 |
| A17152 | F | A1*004, A1*011, B*001, B*066 | No | RP | 13 | 20000 | 4.04E+08 |
| A16187 | M | A1*002, *A1*001*, B*015, B*002 | Yes—1–2 wks of age | RP | 8 | 4000 | 7.29E+07 |
| A16188 | M | A1*004, B*012 | Yes—1–2 wks of age | RP | 15 | IV | 2.55E+08 |
| A17098 | M | A1*008, A1*023, B*012, B*015 | Yes—1–2 wks of age | RP | 13 | 20000 | 1.51E+08 |
| A17099 | F | A1*011, A1*023, B*012, B*024 | Yes—1–2 wks of age | RP | 10 | 12000 | 1.22E+08 |

RP infants had similar levels of CD4 cells pre-SIV infection through early chronic infection, compared to TypP infants. However, CD4 levels were significantly higher in the RP group (mean 3.2 fold increase) during chronic infection (**Fig 3C**). Increased CD4 levels in RP macaques have been observed previously, where it was associated with an expansion of naive T cells [32]. This finding indicates that the infant RP phenotype is not necessarily associated with depletion of total peripheral CD4 T cells.

Chronic activation of CD4 and CD8 T cells is an established correlate of progression to AIDS in both HIV and SIV infections [34–36]. To explore the link between T cell activation and the rapid progression phenotype observed in infant macaques, the levels of HLA-DR were assessed on T cell subsets within both macaque groups (**Fig 4A and 4B**). The level of HLA-DR + CD8 T cells was significantly increased in the TypP at both 4–6 (2.9-fold) and 10–12 (6.6-fold) weeks post-SIV (**Fig 4A**). In the CD4+ population, the TypP macaques exhibited modest yet significant increases in HLA-DR levels at the 4–6 week time point (1.7-fold) (**Fig 4B**). Loss of gut barrier function and associated microbial translocation has been characterized as a significant driver of T cell activation and inflammation during both chronic HIV and SIV infections [35,37]. To evaluate microbial translocation in TypP and RP infants, soluble CD14 (sCD14) and LPS binding protein (LPB) concentrations were measured in plasma collected at weeks 0, 2, 4–6, and 10–12 post-SIV (**S2A and S2B Fig**). These findings demonstrate that despite a more rapid disease progression, the RP infants exhibit similar levels of sCD14 and LPB, as well as low levels of CD4 and CD8 T cell activation.

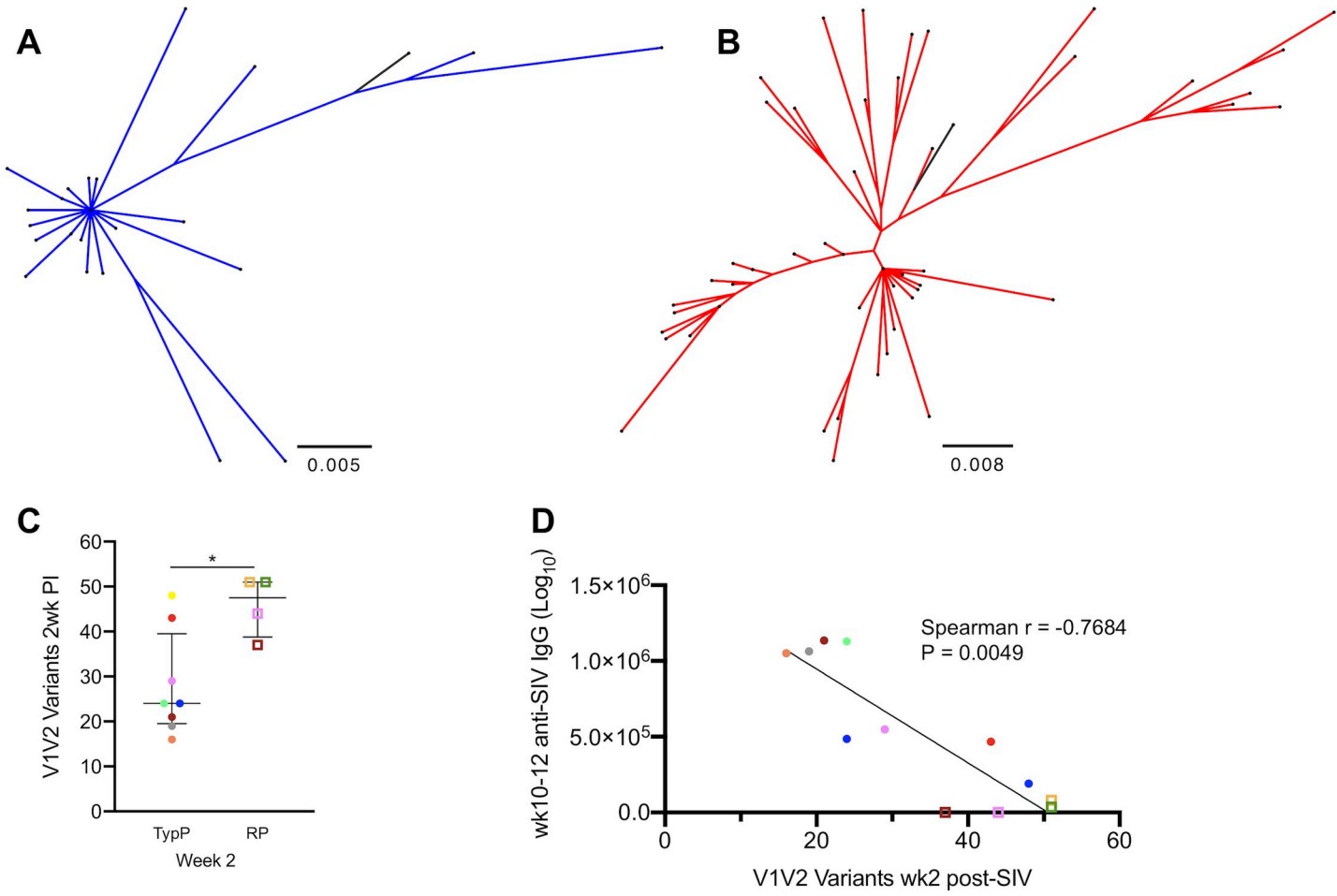

**Fig 2. Number of unique plasma viral variants detected at week 2 post infection negatively correlates with anti-SIV antibodies at wk10.** SIV viral variants were assessed at 2 weeks post-infection from plasma viral RNA on an Illumina MiSeq platform. Representative phylogenies are shown for TypP (**A**) and RP (**B**) infants. The total number of unique plasma viral variants was quantified and compared between TypP (**closed circles**) and RP (**open squares**) infants (**C**). Number of viral variants were compared against plasma anti-SIVenv IgG concentration (**D**). Phylogenies were constructed using maximum likelihood with a GTR substitution model. Black lines in trees represent challenge stock consensus sequence. Statistical tests used to compare infant groups were carried out as described in the methods * = p<0.05. Error bars are shown as either mean with standard deviation or median with interquartile range based on data distribution.

## Rapidly progressing infants fail to increase activated memory B cells during chronic infection

The lack of SIV-specific antibodies in the RP infant macaques raised the question as to whether memory B cell levels and activation were altered in RP compared to the TypP macaques. Assessment of B cells levels within peripheral blood revealed that the percentage of B cells (% CD20+ of CD3- PBMCs) was 1.5-fold higher at 10 to 12 weeks post-SIV in the TypP macaques compared to RP (**Fig 5A**). Bidirectional interactions between T cells and B cells are necessary for effective humoral responses, and ineffective B cell costimulatory function has previously been linked to impaired CD80 expression on B cells of HIV-viremic patients [38]. Comparing CD80+ B cells in RP and TypP macaques revealed that RP infants failed to increase the levels of this cell population during early chronic infection (**Fig 5B**). By 5 weeks post-SIV TypP infants exhibited significantly higher CD80+ B cells than RP infants and this trend continued through 12 weeks post-infection, with TypP infants having on average 8.4-fold higher levels of circulating B cells expressing CD80. To further characterize changes in B cell populations

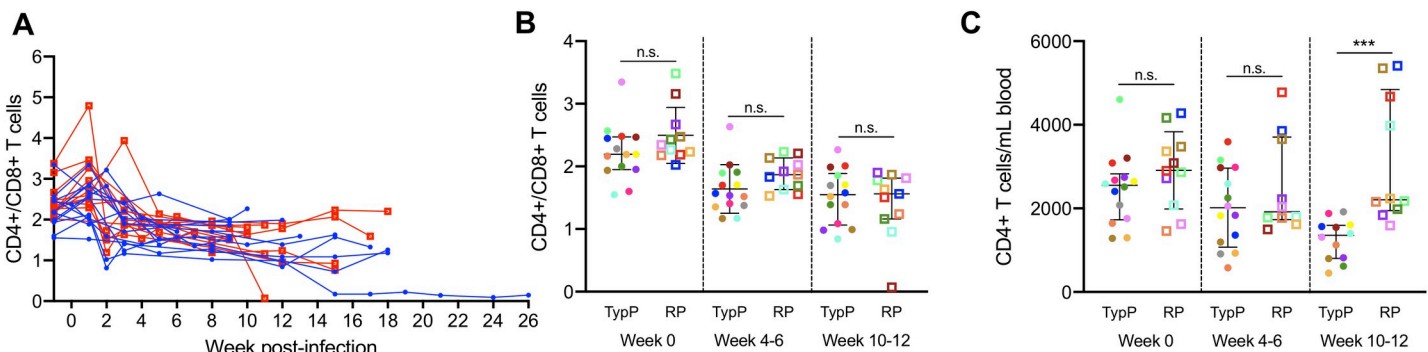

**Fig 3. Total CD4 T cell levels are sustained during chronic infection in RP infants.** CD4 T cell depletion was monitored over time using a CD4/CD8 ratio in PBMC from TypP (**closed circles, blue lines**) and RP (**open squares, red lines**) infants (**A,B**). The total number of peripheral CD4 T cells was monitored following SIV infection (**C**). Statistical tests used to compare infant groups were carried out as described in the methods * = p<0.05, ** = p<0.01, *** = p<0.001. Error bars are shown as either mean with standard deviation or median with interquartile range based on data distribution.

following infection in both groups of infants, B cell memory subsets were evaluated in PBMC based on expression of CD21 and CD27 on CD20+ B cells. TypP infants exhibited elevated levels of activated memory (CD21-, CD27+) B cells compared to the RP macaques (9.8-fold) at 10–12 weeks post-SIV (**Fig 5C**). In addition to an increase in activated memory B cells in TypP infants from baseline (3.3-fold), we observed a significant decrease from baseline levels of activated memory B cells in RP infants (3-fold). The decrease in activated memory B cells in RP infants was offset by an elevation in the level of naive B cells (CD21+, CD27-) (**Fig 5D**). Shifts in proportions of B cell subsets were restricted to naive and activated memory compartments, as no differences were observed in proportions of resting memory (CD21+, CD27+) and tissue-like memory (CD21-, CD27-) B cells (**S3A–S3C Fig**). Previous studies have identified an increase in activated memory B cells expressing CXCR3 and CD11c with chronic viral

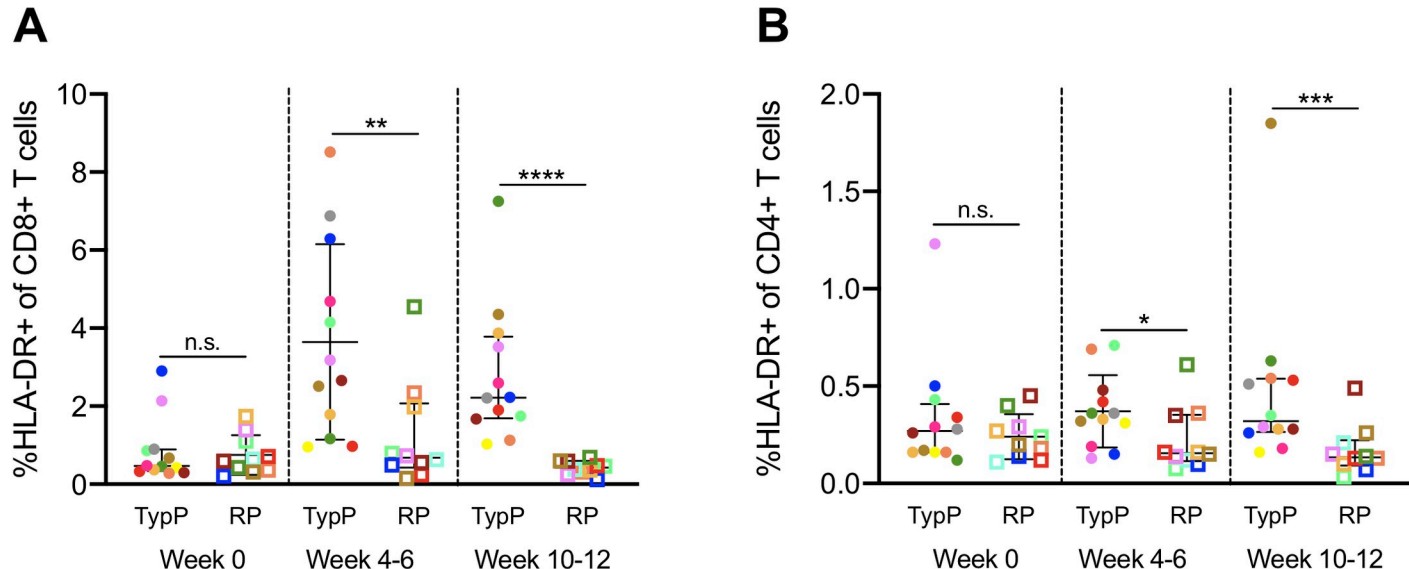

**Fig 4. Activation of CD4 and CD8 T cells is increased in typically progressing infants.** Levels of HLA-DR+ CD4 (**A**) and CD8 (**B**) T cells in peripheral blood were evaluated in TypP (**closed circles**) and RP (**open squares**) infants. Both paired and unpaired and parametric and nonparametric tests were used to compare groups depending on the distribution of the data. ** = p<0.01, *** = p<0.001, **** = p<0.0001. Error bars are shown as either mean with standard deviation or median with interquartile range based on data distribution.

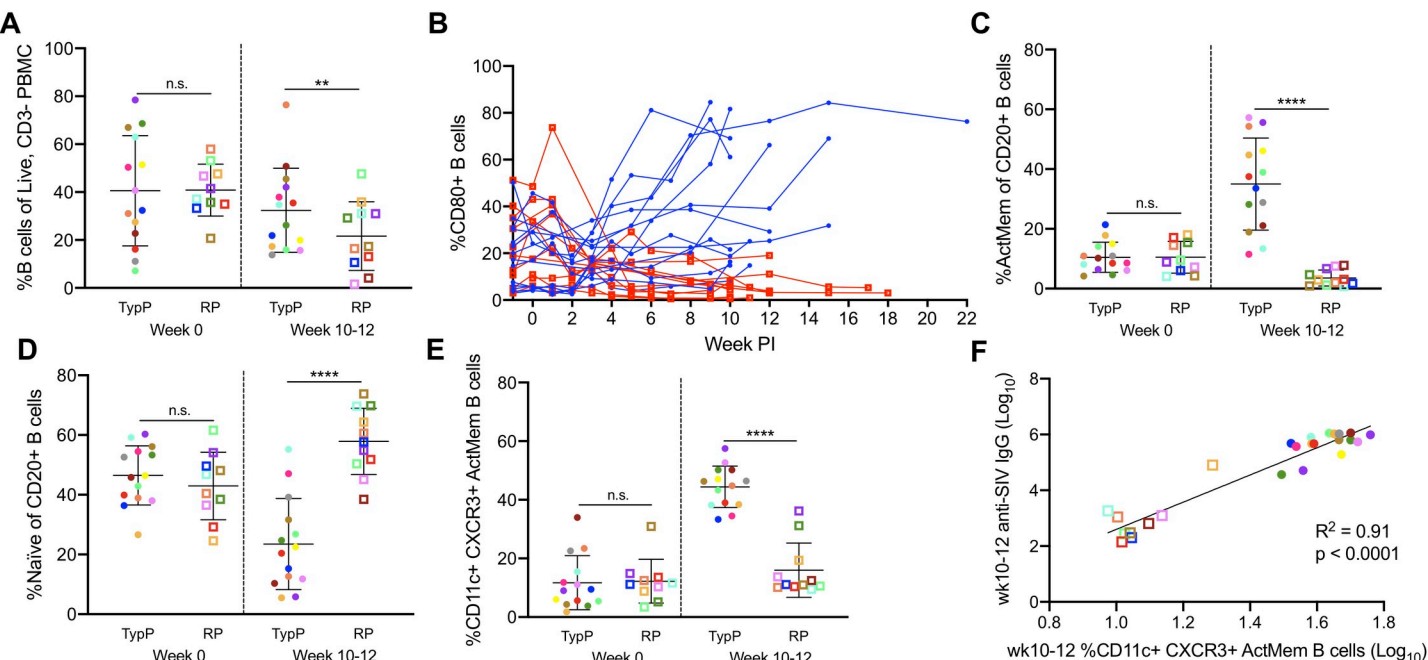

**Fig 5. Rapidly progressing infants fail to increase activated memory B cells during chronic infection.** Proportions of CD20+ B cells (**A**) as well as CD80+ activated B cells (**B**) are shown from PBMC of TypP (**closed circles, blue lines**) and RP (**open squares, red lines**) infants. Proportions of activated memory (**C**) and naïve (**D**) B cell populations from total CD20+ B cells are also compared between TypP and RP infants. Within activated memory B cells we compared atypical CD11c, CXCR3 double positive cells in both TypP and RP infants (**E**) and correlations with anti-SIV antibody levels are shown at week 10–12 (**F**). Statistical tests used to compare infant groups were carried out as described in the methods ** = p<0.01, *** = p<0.001, **** = p<0.0001. Error bars are shown as either mean with standard deviation or median with interquartile range based on data distribution.

infections in mice and humans, as well as with regard to HIV-specific antibody responses [39,40]. Since a defining characteristic of the RP infants is an inability to produce SIV-specific IgG [41], we examined the association between CXCR3+ CD11c+ memory B cells and plasma anti-env IgG concentrations. The proportion of activated memory (CD27+,CD21-) B cells expressing CXCR3 and CD11c was significantly elevated in the TypP compared to the RP infant macaques (**Fig 5E**), and expression of these two markers on activated memory B cells directly correlated with the levels of anti-SIV antibodies present at week 10–12 post-infection (p<0.0001) (**Fig 5F**). These findings provide evidence that low levels of plasma anti-SIV-Env antibodies are due to insufficient memory B cell activation in RP infant macaques.

## Rapidly progressing infants exhibit germinal center dysfunction in secondary lymphoid tissues

Migration of GC B cells as well as Tfh cells into follicles of lymphoid tissues is coordinated by the chemokine CXCL13 [42,43], which is predominantly expressed by follicular dendritic cells and macrophages [42]. Evaluation of plasma CXCL13 identified significantly increased concentrations in the plasma of TypP, but not RP, macaques at week 2 (1.9-fold) and weeks 4–6 (1.5-fold) compared to week 0 (**Fig 6A**). Assessment of Tfh (CD4+ CXCR5+ PD-1hi) within the axillary lymph nodes indicated that by 3–4 weeks post-SIV, TypP infants had significantly higher levels of Tfh cells than RP infants (2.31-fold) (**Fig 6B**). At necropsy, TypP infants experienced a significant increase in Tfh levels from the 4–6 week early chronic timepoint (3.35-fold) while RP infants experienced significant reductions (6.59-fold), resulting in dramatic differences in Tfh cell levels between the 2 groups (>50-fold difference). B cells in the

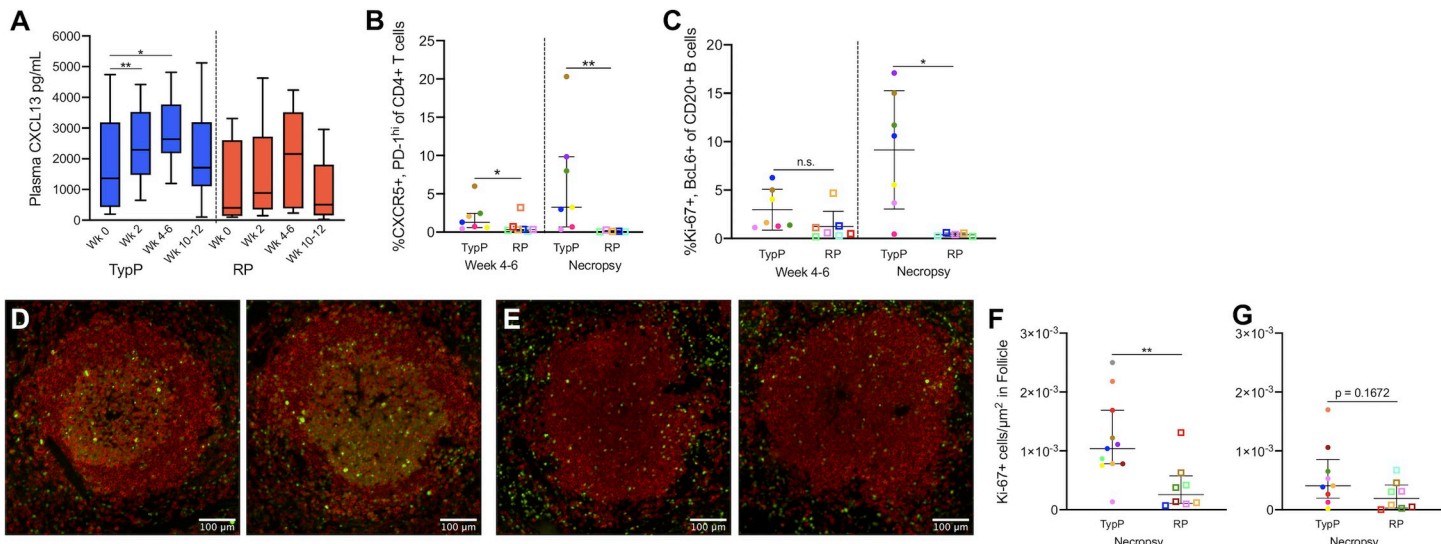

**Fig 6. Rapidly progressing infants exhibit germinal center dysfunction in secondary lymphoid tissues.** Plasma CXCL13 concentrations were measured at timepoints prior to and following SIV infection in TypP (**blue bars**) and RP (**red bars**) infants (**A**) Levels of T follicular helper cells (CXCR5+, PD-1+)(**B**) and germinal center B cells (Ki-67+, BcL6+)(**C**) were measured in axillary lymph node biopsies from early chronic (4–6 weeks post-infection) and necropsy. Paraffin embedded spleens from RP and TypP infants were sectioned and immuno-stained for B cells (CD20, red) proliferation (Ki-67, green) to identify splenic B cell follicles and germinal centers. Representative images are shown for TypP infants (**D**) and RP infants (**E**). Whole sections were scanned and stitched, and Ki-67 positive foci were quantified in B cell follicles of spleen (**F**) and axillary lymph node (**G**). Statistical tests used to compare infant groups were carried out as described in the methods ** = p<0.01. Error bars are shown as either mean with standard deviation or median with interquartile range based on data distribution.

lymph nodes associated with germinal centers (Ki-67+, BCL6+) were similarly elevated in TypP infants at necropsy compared to RP infants (**Fig 6C**). The proportion of GC B cells also underwent an expansion (3.09-fold) in TypP infants while contracting in RP infants (3.12-fold) during the period from early chronic SIV infection until necropsy. This differential outcome resulted in 23-fold higher levels of GC B cells in the TypP infants by the time of necropsy compared to RP infants. To further evaluate the GC B cells (follicular, CD20+, Ki67+) their levels were assessed in lymph nodes and spleen (**Fig 6D and 6E**). Significantly more GC B cells were observed in splenic follicles of TypP versus RP (**Fig 6F**), however levels of GC B cells were similar within the lymph nodes (**Fig 6G**). Together these findings suggest a failure to induce functional germinal centers in spleens of RP infants, while TypP infants undergo a more typical expansion of Tfh and GC B cells.

## Elevated interferon-induced immune activation in lymphoid tissues and B cell follicles of RP infants

Type-I interferon associated immune changes have previously been described as a significant factor driving pathogenesis of HIV and SIV [27–29,44]. Assessment of plasma IFNα identified sustained elevated levels within the RP macaque plasma compared to the TypP macaques at both 4–6 and 10–12 weeks post-infection, with 8-fold higher concentrations of IFNα in RP infants (mean 270pg/ml) compared to TyP (33 pg/ml) (**Fig 7A**). During activation and maturation, pDCs have been shown to increase expression of the costimulatory marker CD80 [45,46]. Therefore, pDC activation was measured as the proportion of CD80+ pDC out of total CD123+/CD11c- CD14- cells using flow cytometry. We determined that while there were similar levels of CD80 on circulating pDC, these levels were significantly higher in the axillary lymph nodes of RP macaques, with the proportion of CD80+ pDC in RP infants doubling that

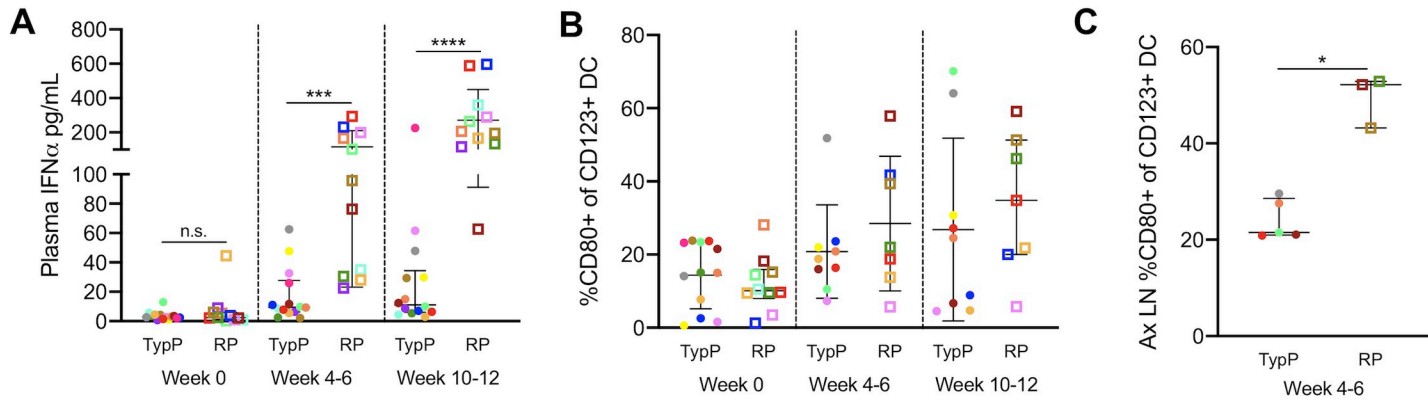

**Fig 7. Elevated Interferon-α and pDC activation in RP infants.** Plasma IFNα concentrations were measured by ELISA at multiple time points in TypP and RP infants (**A**). Proportions of activated CD80+ plasmacytoid dendritic (CD123+) in PBMC from TypP (**closed circles**) and RP (**open squares**) infants were evaluated in blood (**B**) and lymph nodes (**C**) by flow cytometry. Statistical tests used to compare infant groups were carried out as described in the methods * = p<0.05, *** = p<0.001, **** = p<0.0001. Error bars are shown as either mean with standard deviation or median with interquartile range based on data distribution.

of TypP infants by 4–6 weeks post-SIV (**Fig 7B and 7C**). This finding identifies activated pDC in lymphoid tissues as a potential source of elevated plasma IFNα in RP macaques.

To assess a direct association between elevated levels of type 1 IFN and the inability to produce high levels of SIV-specific antibodies the expression of the IFN-induced protein MX1 was evaluated in splenic B cell follicles. While MX1 was detected in the extrafollicular area of both TypP and RP macaques (**Fig 8A and 8D**), relatively low levels of follicular MX1 production were observed in TypP infants (**Fig 8B and 8C**). In contrast, RP macaques exhibited significantly increased MX1 expression in splenic germinal centers (**Fig 8D–8G**). Regions of elevated MX1 in RP infants which corresponded to sites proximal to follicles were identified as expressing a majority of SIV-infected cells by RNA-scope *in situ* hybridization (**Fig 9A and 9B**). Scanning entire splenic sections determined that RP infants had significantly more SIV-positive cells/mm$^2$ compared to TypP infants, and that these cells are largely present in the T cell zones proximal to B cell follicles (**Fig 9C**). These data demonstrate that increased IFN responses are observed in sites of B cell maturation in RP macaques, and that this is associated with an inability of RP infant macaques to mount SIV-specific antibody responses.

## Discussion

Factors contributing to rapid progression to AIDS in infants remain poorly understood. Here, we demonstrate that 44% (11/25) of infant macaques infected with SIVmac251 between 6 and 18 months of age develop elevated SIV replication, very low or undetectable levels of SIV-specific antibodies and a more rapid disease course. Importantly, the frequency, as well as immunological, virologic, and clinical aspects of this phenotype recapitulate what has been reported in rapidly progressing HIV-infected infants [2,4,7,8], and thus our findings likely reflect underlying factors driving more severe clinical outcomes. An evaluation of the immune dysfunction observed in RP infant macaques provides evidence for altered memory B cell and T follicular helper cell levels, as well as activation of lymphoid pDCs. Importantly, we have demonstrated an elevated type-I IFN-induced protein expression, similar to previous studies of rapidly progressing macaques [25,26]. The elevated type-1 IFN was observed in B cell follicles and was associated with GC dysfunction, supporting the hypothesis that aberrant type I IFN production contributes to B cell dysfunction and failure to mount humoral responses in rapidly progressing infants. An interesting finding from this study was a lack of evidence that RP

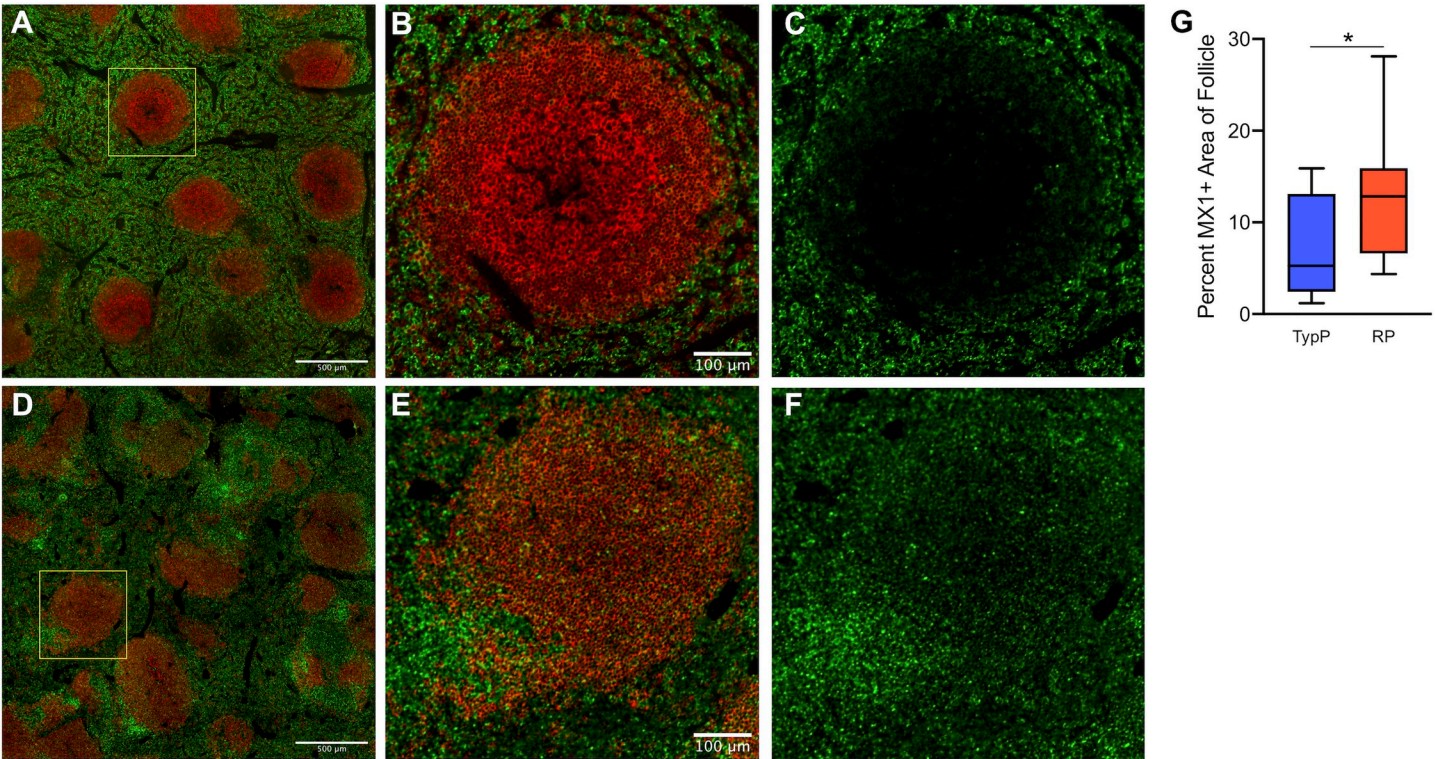

**Fig 8. Increased type 1 IFN associated protein expression is observed in B cell follicles of RP infants.** Levels of MX1 protein were measured in areas within B cell follicles of spleen. Paraffin embedded spleens from RP and TypP infants were sectioned and immuno-stained for B cells (CD20, red) and MX1 (green) to identify splenic B cell follicles and IFN-induced protein expression. Representative images are shown for TypP infants (**A-C**) and RP infants (**D-F**). Whole sections were scanned and stitched and MX1 was quantified within 10 randomly selected splenic B cell follicles (**G**) for TypP (**blue bar**) and RP (**red bar**) infants. Statistical tests used to compare infant groups were carried out as described in the methods * = p<0.05.

phenotype in infant macaques was associated with increased T cell activation or microbial translocation, as has been observed previously [35–37,47]. In contrast, TypP infants exhibited moderate levels of SIV plasma viremia, CD4 T cell depletion, systemic immune activation and development of hyperplastic B cell follicles more routinely associated with progression to simian AIDS [30,36].

Identification of multiple genotypes at the initiation of infection (week 2) can be due to two possible explanations. First, there may be an increase in the number of founder viruses that infect the macaques via the oral route in the RP compared to the TypP infant macaques. Second, it is possible that increased V1V2 variants is the result of increased viral replication, and accompanying genetic mutations, due to factors intrinsic to the RP infants. Previous work from Tsai et al identified a link between rapid SIV disease progression and the number as well as relatedness of viral variants determined by sequencing of SIV*env* gene [33]. This study used an R5 SHIV to infect adult female macaques intravaginally and identified a subset of macaques that failed to control viremia and developed AIDS at 30 weeks post-infection. Similar to our findings, this study found that rapidly progressing macaques had a greater number of plasma viral variants, with fewer genetic variants identified in the more typical "chronic progressors".

Our finding raises questions with regard to what is driving this differential Tfh levels in TypP and RP macaques. While there is evidence for both expansion as well as early depletion of Tfh cells during SIV infection [48,49], it is poorly understood whether loss of Tfh cells, and their precursors, may be attributed to direct killing from SIV or to aberrant inflammatory

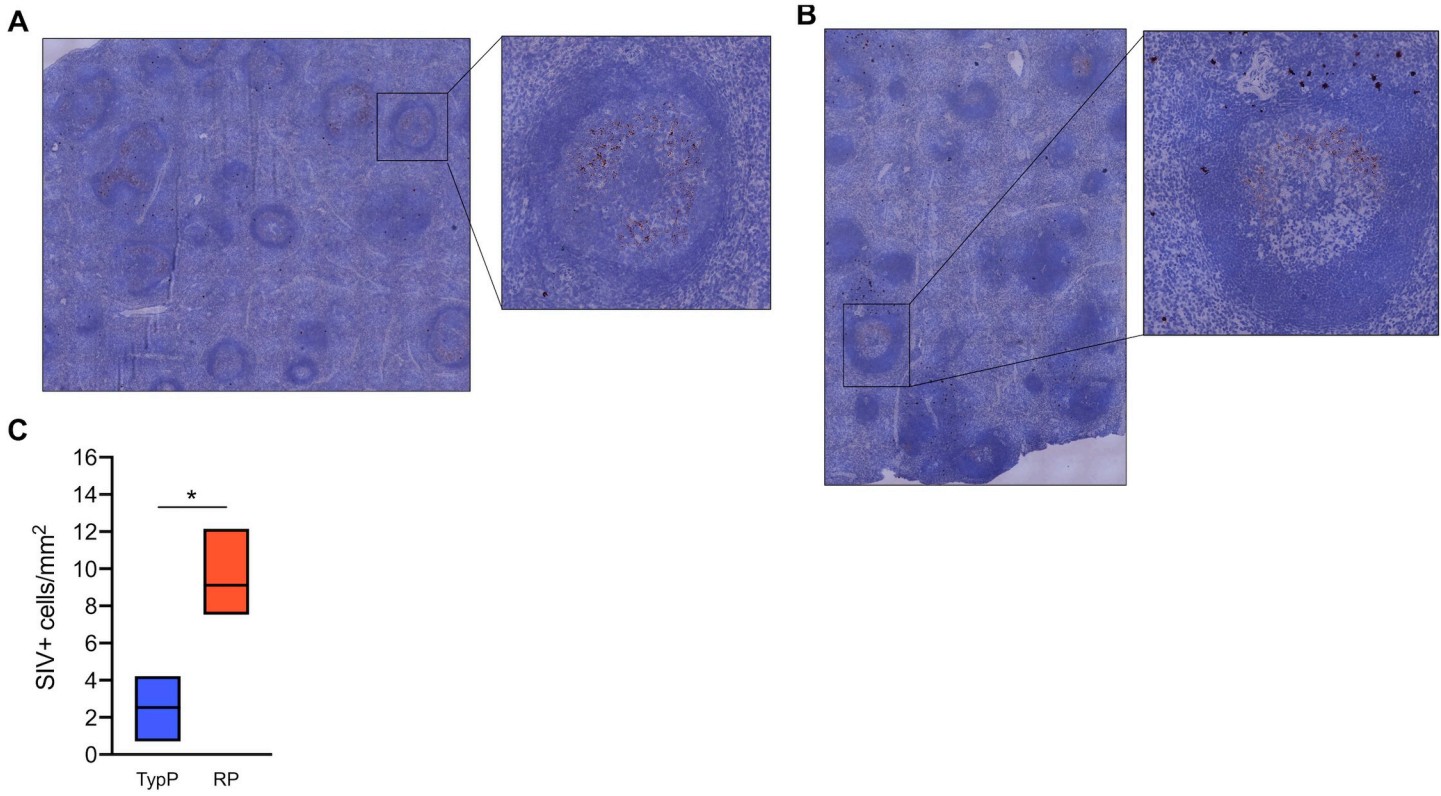

**Fig 9. Rapidly progressing infants have more SIV infected cells localized outside of splenic germinal centers.** RNAscope in situ hybridization using SIV-specific RNA probes was used to detect SIV-infected cells and cell-free virus in the spleens of TypP (**A**) and RP (**B**) infants. SIV+ cells were quantified across stitched images and normalized for area of splenic tissue (**C**). Statistical tests used to compare infant groups were carried out as described in the methods * = p<0.05. Error bars are shown as floating bars (min to max) with line indicating the mean. n = 3 per group.

signaling driving apoptosis [49,50]. A 2008 study by Cumont et al provides evidence for increased lymphoid cell apoptosis contributing to rapid disease progression in Indian rhesus macaques with low levels of apoptosis in primate species that exhibit a controlled or lack of disease progression [51]. More recently, it was demonstrated that scattered expression of lymphoid CXCL13 is associated with an impaired follicular architecture and reduced homing of Tfh cells to B cell follicles [52]. Thus, reduced Tfh in RP infants, and associated reductions in active germinal centers, may also be attributed to spatially altered CXCL13 expression impaired homing of T cells into B cell follicles. Specific infection of Tfh cells by HIV and SIV has been observed in both adult and infant macaques [53–56], yet rather than direct infection driving Tfh loss, infected Tfh cells are reported to serve as sanctuaries for viral persistence [53]. While RNAscope identified similar levels of cell free virus within B cell follicle light zones between RP and TyP infants (**Fig 9**), the majority of SIV-infected cells, as well as the brightest areas of MX1 staining, were observed on the border of the B cell follicle, which can be populated by Tfh as well as other CD4 T cells. It is therefore likely that a combination of SIV associated cell killing and IFN-induction of potentially proapoptotic genes [57–61] may be playing roles in the Tfh loss observed in the RP infant macaques.

These data allow for the elucidation of a model that summarizes our findings and outlines factors that may influence the RP phenotype outcome in the SIV infant macaques (**Fig 10**). Temporally our first observation was the increase in genetic variability in SIV env V1V2 at 2 weeks post-infection (**Section in A Fig 10**). This may be a contributing factor to, or conversely

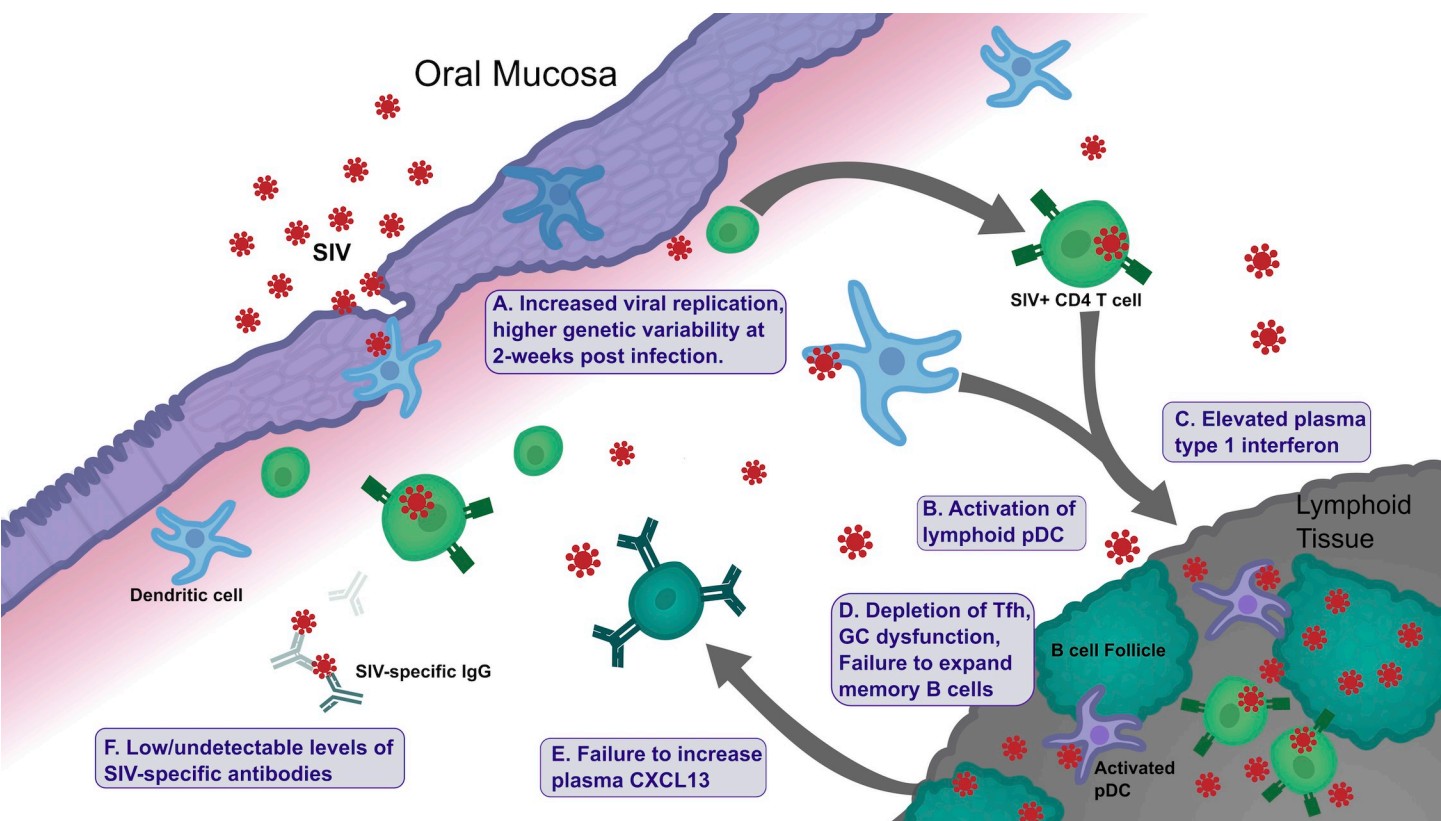

**Fig 10. Model summarizing factors influencing rapid SIV progression in infants** Illustration of events following infection in infant macaques exhibiting rapid disease progression. GC = germinal center; Tfh = T follicular helper cell; pDC = plasmacytoid dendritic cell.

a consequence of, progression toward the RP phenotype. Viral replication likely drives activation of pDCs in tissue, including lymph nodes where they were observed at 4–6 weeks post-infection (**Section in B Fig 10**), The pDC activation is the likely source of elevated type 1 IFN production, which was observed in the plasma at that same timepoint (**Section in C Fig 10**). The increased pDC activation in the RP infants is in agreement with a previous study identifying increased lymphoid pDC number in non-progressors [20], and supports a role for lymphoid pDCs either being stimulated by or contributing to the pathogenicity of SIV infection in infant macaque. The prolonged elevated type 1 IFN could have direct effects on expansion of memory B cells and B cell function [61–63], as well as potentially influencing Tfh cell function within the lymph nodes [64] (**Section in D Fig 10**). Germinal center dysfunction may also be influenced by failing to increase concentrations of the chemokine CXCL13 (**Section E in Fig 10**). This GC dysfunction is associated with a failure to mount an effective anti-SIV antibody response (**Section F in Fig 10**). The findings described here therefore provide insights into the early response to the SIV infection driving outcomes toward either rapid or typical disease progression.

To summarize, these findings provide an in-depth characterization linking virologic and immunological aspects of rapid SIV progression in infant macaques. Moreover, rapidly progressing macaques described here display a distinct pathology defined by failure to mount humoral immune responses and reflects key aspects of rapid progression to AIDS in infant humans. Our results also provide a detailed description of germinal center dysfunction and point to a role for aberrant IFN-signaling in germinal centers as a potential driver of this

outcome. While survival of HIV-infected infants has improved proportionally with access to antiviral therapy, early severe disease still occurs and often precedes antiretroviral therapy [65]. These findings build on our understanding of infant HIV pathogenesis and can potentially be used towards the development of improved therapies and interventions.

## Methods

### Ethics statement

All animal studies were approved by the University of Washington Animal Care and Use Committee (IACUC) under protocol #4213. Infant macaques were housed in the specialized infant care wing of the Washington National Primate Research Center (WaNPRC). Infants were fed infant formula and soaked standard monkey biscuits supplemented with fruits and vegetables. Water was provided ad libitum. Enrichment material and surrogates were provided for animal well-being and infants were allowed several hours a day for socialization in groups of four. Anesthesia was administered during sample collection as either ketamine hydrochloride or zolazepan (Telazol), based on samples collected. Doses were calculated from daily weight measurements. Animals showing significant distress attributed to SIV infection were euthanized between 14- and 17-weeks post-infection. To further minimize animal suffering, a study endpoint of 10 weeks post infection was added to the IACUC protocol. Animals were euthanized by humanely administering ketamine hydrochloride followed by an overdose of barbiturates according to the guidelines of the American Veterinary Medical Association.

### Study animals

25 infant rhesus macaques were purchased and transported from the Oregon National Primate to WaNPRC between 2 and 5 weeks of age. At 5–10 weeks of age, up to 8 oral SIV challenges were administered as escalating doses ranging from 1000 TCID50 to 20000 TCID50 of SIV-mac251. SIVmac251 challenges were prepared from rhesus macaque PBMC-grown supernatant, obtained from NIH AIDS Reagent Program and Dr. Nancy Miller (NIH/NIAID). Virus was diluted to 0.25 mL in RPMI1640 media and delivered dropwise across the oral mucosa via needleless syringe. Infection was confirmed by plasma viral loads (WaNPRC Virology Core). Infants that remained uninfected (plasma viral load ≤30 copies SIV RNA/mL for 2 weeks post-challenge) after 8 oral challenges were infected intravenously with 500 TCID50 of the same challenge stock (3 macaques were infected in this way) to evaluate disease pathogenesis in all infants. Infants were monitored for 9 to 12 weeks after SIV infection before being euthanized. MHC-I haplotype analysis was carried out by the University of Wisconsin Madison Genetic Services Unit. Rhesus MHC-1 typing did not account for the frequency of rapid progression between MHC genotype and skewed susceptibility to SIV in either study group (Table 1).

### Phenotypic analysis of immune cell subsets

Phenotypic analyses of PBMC and lymph node cell populations were performed by multiparametric flow cytometry. Freshly isolated PBMC were counted and stained as previously described [66]. Briefly, activation of T cells (CD3+, CD20-, CD14-), classical monocytes (HLA-DR+, CD14+, CD16-, CD20-, CD3-) and CD16+ monocytes (HLA-DR+, CD14+, CD16+, CD20-, CD3-) was assessed using gating strategies outlined in **S1 and S2 Figs.** The following antibodies were used: CD3(SP34-2)-Pacific Blue and APC, CD4(OKT4)-BV650 and APC-Cy7, CD8(SK1)-APC-H7, CD20(2H7)-BV570 and PE, CD14(M5E2)-BV785 and APC-H7, CD16(3G8)-BV605, CD11c(S-HCL-3)-APC, CD123(7G3)-PerCP-Cy5.5, CCR5

(3A9)-APC, CXCR3(1C6)-PE-CF594, CD38(AT-1)-FITC, Ki-67(B65)-PE and FITC, HLA-DR (L243)-BV711 and PE, CD80(L307.4)-PE-Cy7, CD83(HB15e)-PE-CF594, CD86(2331)-BV711 (BD Life Sciences). Stained cells were washed and fixed in 1% paraformaldehyde before analysis on a LSR-II flow cytometer (BD Biosciences). Compensation and analysis were performed using FlowJo version 10 (v. 10.5.3, FlowJo LLC).

## Analysis of plasma protein and IgG/IgA concentrations

Plasma samples were collected after centrifugation of whole blood collected in EDTA tubes. Plasma IFNα concentrations were measured with the Human IFN-α ELISA$^{PRO}$ kit (Mabtech), following the manufacturer's instructions. Plasma CXCL13 was measured using a Human BLC ELISA kit (Ray Biotech). Plasma sCD14 was measured using the Human CD14 Quanti-kine ELISA kit (R&D Systems). Plasma LPS binding protein (LBP) was measured using an LBP quantification immunoassay (Biometec). Measurement of plasma anti-SIVgp140 IgG and IgA concentrations was performed using a custom ELISA containing wells coated with rGP130 (NIH-ARP #12797). Samples were serially diluted and absorbance values were fit to standard curves generated using either purified rhesus IgG or IgA.

## SIV env sequence analysis

Viral RNA was isolated from rhesus macaque plasma from 2 weeks post-SIV infection using an Ultrasense Viral RNA kit (QIAGEN) and cDNA was reverse transcribed using the Applied Biosystems High Capacity cDNA synthesis kit (Thermo-Fisher). Libraries were prepared using a 2 step PCR protocol for amplifying the 584bp product with adaptors. First round primers: F: TAGAGGATGTATGGCAACTC and R: CTTGTGCATGAAGAGACCA. Second round primers: F: TCGTCGGCAGCGTCAGATGTGTATAAGAGACAGTATGGCAACTCTTTGA GACC and R: GTCTCGTGGGCTCGGAGATGTGTATAAGAGACAGGAAGAGACCACC ACCTTAG. PCR products were FLASH-gel purified and a 5-cycle indexing PCR was used for addition of P5 and P7 Illumina indices. Libraries were loaded onto an Illumina 600 cycle V3 cartridge according to the manufacturer's instructions and run on an Illumina MiSeq as described previously [67]. Amplicons were reconstructed from forward and reverse FASTQ reads via FLASH with maximum and minimum parameters set so that >95% of reads were aligned. Adaptor sequences were removed using cutadapt with error rate of 0.3 and minimum length of 100bp. Sequence quality filters were applied using FASTQ_quality_filter with a minimum quality score of 20, and the minimum percent of bases that must have the set minimum quality score to 100%. Duplicates were removed using dedupe2.sh, allowing for maximum mismatches of 5 and maximum edits set to 2. In individual plasma samples from infants infected by oral challenge, this approach provided an average of 258 thousand sequences after filtering paired reads based on quality score. After collapsing identical sequences, unique variant clusters were identified as a set of sequences with no more than 5bp mismatches among them. Additionally, these unique variants were present at levels of at least 10 copies to be considered for further analysis. Fastq files were converted to fasta and aligned with Clustal Ω and unique genotypes were enumerated. Alignments were uploaded to the DIVEIN analysis server where phylogenies were constructed by maximum likelihood using the generalized time reversal substitution model [68]. Pairwise distance was calculated using a consensus sequence derived from the 331 sequenced V1V2 variants represented within our SIVmac251 challenge stock.

## Tissue imaging

Immunofluorescence microscopy was carried out as previously described [69], with the following exceptions: B cells were targeted using α-CD20 (clone EP459Y, Abcam, 1:300);

proliferating cells were targeted using anti-Ki-67 (clone MM1, Leica Biosystems, 1:100); cells responding to IFN signaling were targeted by anti-Mx1 (clone M143, EMD Millipore, 1:500). CD20 was detected using goat anti-rabbit Alexa Fluor 594 (Life Technologies, 1:500) while Ki-67 and Mx1 were detected using goat anti-mouse Alexa Fluor 488 (Life Technologies, 1:500). Spleen and lymph node and sections were scanned under 100x magnification using a Nikon Eclipse Ti inverted fluorescent microscope (Nikon, Melville, NY) to capture and stitch multiple fields. Analysis was carried out on ten randomly selected B cell follicles per tissue. Quantification of Ki-67 was carried out in Imaris software by manually selecting follicles followed by counting Ki-67 using the spots tool. Mx1 was quantified in Fiji by selecting B cell follicles and thresholding a mask using the "Moments" parameters followed by counting the number of Mx1 positive and Mx1 negative pixels to calculate percent of pixels positive for Mx1.

In situ hybridization analysis of SIV RNA in the spleen was carried out as previously described [69]. Assays were carried out using RNAscope technology (Advanced Cell Diagnostics). Spleen sections (5μm) on glass slides were baked at 60˚C for 1 hour before deparaffinization in xylene (2 × 5 minutes) followed by 100% ethanol (2 × 3 minutes). Slides were then pretreated with hydrogen peroxide reagent to quench endogenous peroxidases. Antigen retrieval was performed by boiling slides for 20 minutes in antigen retrieval buffer followed by washing in deionized water and ethanol and baking for 30 minutes at 60˚C. SIVmac239 probes (Advanced Cell Diagnostics) targeting SIV gag, pol, tat, rev, env, vpx, vpr, nef and rev genes were then hybridized to tissue for 2 hours at 40˚C. Following the recommended six amplification steps, DAB-A and B reagents were mixed and incubated with tissue until visual detection of brown color was achieved. Tissues were counterstained with CAT Hematoxylin for 30 seconds and briefly rinsed with tap $H_2O$. Coverslips were mounted using Permount mounting media and allowed to cure overnight before imaging. Scanned images of whole mounted tissue cross sections were acquired as described above. SIV+ cells were quantified in FIJI software by color thresholding followed by the particle analysis tool with size parameters adjusted to detect only cells.

## Statistical analysis

All statistical analyses were performed using either Prism v.8 (Graph Pad) or R version 3.5.1. Data distributions were assessed using D'Agostino and Pearson normality tests. Comparisons of proportions of immune cell populations and cytokines across and between TypP and RP infants were made using 2-tailed Mann–Whitney U tests, Wilcoxen matched-pairs signed-ranks tests, or t-tests when appropriate.

## Supporting information

**S1 Fig. Viral variants from RP infants are represented across more diverse challenge stock lineages.** Phylogeny of macaque-derived plasma V1V2 variants isolated at 2 weeks post-infection(colored) and variants within challenge stock (black) (**A**). The number of clades in which each macaque-derived variant was represented was enumerated and the numbers of representative clades were compared between variants from TypP and RP macaques (**B**). Phylogenies were constructed using maximum likelihood with a GTR substitution model using the challenge stock consensus sequence. Statistical tests used to compare infant groups were carried out as described in the methods * = p<0.05. Error bars are shown as either mean with standard deviation or median with interquartile range based on data distribution.
(TIF)

**S2 Fig. Similar innate responses to microbial products are observed in typical and rapidly progressing infant macaques.** Plasma sCD14(**A**) and LBP(**B**) concentrations were measured by ELISA at timepoints prior to and following SIV infection in TypP and RP infants. Statistical tests used to compare infant groups were carried out as described in the methods * = p<0.05. (TIF)

**S3 Fig. No differences are observed in levels of Resting Memory B cells, Tissue-Like Memory B cells, or IL-21R+ Activated Memory B cells in RP and TypP infants.** Proportions of memory B cell subsets in TypP infants (**closed circles**) and RP infants (**open squares**). Percentages of resting memory B cells (RestMem, CD21+, CD27+) and Tissue-Like memory B cells (TLMem, CD21-,CD27-) were evaluated within the total B cell (CD20+) population in PMBC (**A+B**). The percentage of Activated Memory B cells (ActMem, CD21-, CD27+) was measured expressing the IL21 receptor (IL-21R) (**C**). Statistical tests used to compare infant groups were carried out as described in the methods. Error bars are shown as either mean with standard deviation or median with interquartile range based on data distribution. (TIF)

## Acknowledgments

The authors would also like to acknowledge the veterinary and support staff at the Washington National Primate Research Center, Brian Johnson at the University of Washington Histology and Imaging Core for expertise and technical assistance and Roger Wiseman and Eileen Maher at the University of Wisconsin Madison and the Wisconsin National Primate Research Center for assistance with MHC typing.

## Author Contributions

**Conceptualization:** Matthew P. Wood, Ann Chahroudi, Deborah H. Fuller, Donald L. Sodora.

**Data curation:** Matthew P. Wood, Chloe I. Jones, Brian G. Oliver, Katherine A. Fancher.

**Formal analysis:** Matthew P. Wood, Chloe I. Jones, Katherine A. Fancher, James T. Fuller.

**Funding acquisition:** Deborah H. Fuller, Donald L. Sodora.

**Investigation:** Matthew P. Wood, Chloe I. Jones, Adriana Lippy, Brynn Walund, Katherine A. Fancher, Bridget S. Fisher, Piper J. Wright, James T. Fuller, Patience Murapa, Jakob Habib, Maud Mavigner.

**Methodology:** Matthew P. Wood, Chloe I. Jones, Brian G. Oliver, Brynn Walund, Maud Mavigner.

**Project administration:** Matthew P. Wood, Ann Chahroudi, D. Noah Sather, Donald L. Sodora.

**Resources:** D. Noah Sather, Deborah H. Fuller.

**Software:** Matthew P. Wood, Katherine A. Fancher.

**Supervision:** Matthew P. Wood, Ann Chahroudi, Donald L. Sodora.

**Validation:** Matthew P. Wood, Brian G. Oliver, Maud Mavigner.

**Visualization:** Matthew P. Wood, Chloe I. Jones, Donald L. Sodora.

**Writing – original draft:** Matthew P. Wood, Chloe I. Jones, Donald L. Sodora.

**Writing – review & editing:** Matthew P. Wood, Katherine A. Fancher, Deborah H. Fuller, Donald L. Sodora.

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
