## [Decision Letter · Decision Letter 0]

4 Feb 2021

Dear Dr. Sodora,

Thank you very much for submitting your manuscript "Rapid progression is associated with lymphoid follicle dysfunction in SIV-infected infant rhesus macaques." for consideration at PLOS Pathogens. As with all papers reviewed by the journal, your manuscript was reviewed by members of the editorial board and by several independent reviewers. The reviewers appreciated the attention to an important topic. Based on the reviews, we are likely to accept this manuscript for publication, providing that you modify the manuscript according to the review recommendations.

Sincerely,

Daniel C. Douek

Associate Editor

PLOS Pathogens

Richard Koup

Section Editor

PLOS Pathogens

Kasturi Haldar

Editor-in-Chief

PLOS Pathogens

orcid.org/0000-0001-5065-158X

Michael Malim

Editor-in-Chief

PLOS Pathogens

orcid.org/0000-0002-7699-2064

Reviewer Comments (if any, and for reference):

Reviewer's Responses to Questions

**Part I - Summary**

Reviewer #1: This is an interesting report related to infant SIV-infected monkeys addressing the role of immune factors that are associated with disease progression in infants.

This manuscript established several immune parameters associated with rapid progression in infant monkeys.

Therefore, the topic on infants and the results proposed are of interest for Plos Pathogens publication.

Reviewer #2: This excellent paper submitted by Sodora et al. describe thorough investigations into the causes of rapid disease progression in SIVmac251-infected infant rhesus macaques.

The study is performed by a stellar team of senior investigators with outstanding expertise in SIV pathogenesis and vaccines in macaques of young age (Sodora, Fueller, Chahroudi and Sather).

The paper is very well written and describes a series of logic experiments that assess in detail the SIV pathogenesis in infant rhesus macaques with the aim to define the immunologic and virologic profiles of rapid progressors. The study is significant and therefore of high interest for the HIV field and for the PLOS Path readers and the information generated will advance the field. The methods applied are appropriate to answer the scientific question, and the large number of animals allow strong statistical analyses and conclusions.

**Part II – Major Issues: Key Experiments Required for Acceptance**

Reviewer #1: no major issues

Reviewer #2: (No Response)

**Part III – Minor Issues: Editorial and Data Presentation Modifications**

Reviewer #1: My concerns are:

In the abstract the authors indicated “had similar CD4 T Cell depletion”, however this is not supported by figure 3 panel C, please rephrase the sentence.

In the introduction, lanes 74-76 the authors mentioned the relationship between viremia and type I IFN, a reference should be also included (Campillo-Gimenez L, J Virol. 2010 Feb;84(4):1838-46. doi: 10.1128/JVI.01496-09). Indeed, a clear relationship between pDC, IFN levels and monkey pathogenesis is described in this manuscript.

Lane, 77-78, please also included these two references that demonstrated the link between the extent of cell death, B cell follicle activation and abortive Ig responses (Monceaux V, AIDS. 2003 Jul 25;17(11):1585-96. doi: 10.1097/00002030-200307250-00002.; Cumont MC, J Virol. 2008 Feb;82(3):1175-84. doi: 10.1128/JVI.00450-07). These could be also introduced in the discussion and are in line with the observations made in this manuscript.

Lane 79-80 please add “infant” in the sentence.

The authors proposed that the absence of depletion is related to an expansion of naive T cells (lane 144). If the authors have the opportunity to perform additional staining, this will greatly improve the sentence. Thus, the term “unexpectedly” is not appropriate in this context (lane 161).

Lane 221, the authors assessed the link between pDC and IFN in infant monkeys that perfectly fit with the notion described by Campillo-Gimenez L (J Virol 2010).

In the discussion, the authors stated that “aberrant IFN-driven immune activation contributes to B cell dysfunction”. However, there is no immune activation in RP whereas IFN levels were higher. The sentence should be deleted whereas the observations are consistent with previous reports. Thus, in lane 266, the authors should discuss these manuscripts (Campillo-Gimenez L and Cumont) since they are in line with their observations.

Lanes 286-291, In fact Tfh cells are localized at the border of B cell follicle (see Moukambi Plos pathogens and Moukambi F, Mucosal Immunol. 2019 Jul;12(4):1038-1054. doi: 10.1038/s41385-019-0174-0), and perfectly fit with the observations of viral replicative cells and MX1 expression in infant monkeys. Furthermore, in this manuscript (Mucosal immunology), the authors also addressed the expression of CXCL13 in relationship with follicle architecture and therefore should be discussed. Indeed, instead to do a list of results (lane 302-307) this part in infant merits to be further discussed.

Reviewer #2: 1. Lack of seropositivity for antibodies against the virus envelope in rapid progressors SIV-infected nonhuman primates was previously described in several other studies (Ex Pauza et al., Brocca-Cofano et al. etc) and this should be cited and discussed in the current paper. This will strengthen the findings in the current study, as it seems that lack of antibody production is the strongest criteria to differentiate the animal groups in RP and TypP. The survival of the two groups is not used to differentiate the two groups. It looks like the lack of antibody production is a common feature for rapid progression in both young and adult SIV-infected nonhuman primates, belonging to different species and this was reported by several studies.

2. The possibility that the antibodies may in fact be produced in the rapid progressors but complexed by the higher levels of antigens produced by the rapid progressor and therefore may be undetectable in this subgroup should be also discussed.

3. Is there a significant difference in survival between the two groups?

4. The graphs that include all the animals (Ex SIV RNA, ratio CD4/CD8, % of CD80+ B cells would benefit by the usage of just two colors, one for RP and one for the TypP). It is very difficult to differentiate the two groups.

5. Fig. 3 The number of animals in the CD4 graphs (Fig 3 c) is smaller than in Fig 3A and B. Was only a subset of animals used in this?

6. The authors mention a possible difference in the number of viral variants that infect RP and TypP and this may be pursued experimentally (through SGA) to get a definite answer. I understand, however, that the volume of samples taken from the young animals is very small and may have prevented the group from performing these experiments, as they measured numerous other parameters

7. Fig. 8. The detail selected shows MX1 increase adjacent to the germinal center, so the MX1 expression and localization this should be described as “in proximity to the germinal center” instead of “in the germinal center”.

8. Was there any MHC Class I allele responsible for the clinical outcome. From Table 1 it looks like the authors genotyped the animals but they do not comment on this.

PLOS authors have the option to publish the peer review history of their article (what does this mean?). If published, this will include your full peer review and any attached files.

Reviewer #1: **Yes: **Estaquier Jérôme

Reviewer #2: No
---

## [Decision Letter · Decision Letter 1]

20 Apr 2021

Dear Dr. Sodora,

We are pleased to inform you that your manuscript 'Rapid progression is associated with lymphoid follicle dysfunction in SIV-infected infant rhesus macaques.' has been provisionally accepted for publication in PLOS Pathogens.

Best regards,

Daniel C. Douek

Associate Editor

PLOS Pathogens

Richard Koup

Section Editor

PLOS Pathogens

Kasturi Haldar

Editor-in-Chief

PLOS Pathogens

orcid.org/0000-0001-5065-158X

Michael Malim

Editor-in-Chief

PLOS Pathogens

orcid.org/0000-0002-7699-2064

Reviewer Comments (if any, and for reference):

Reviewer's Responses to Questions

**Part I - Summary**

Reviewer #1: The authors provided a response to my concerns.

Reviewer #2: The authors carefully addressed all my previous minor concerns. The manuscripts is improved. This is an excellent study that will be of high interest for the readers of this journal.

**Part II – Major Issues: Key Experiments Required for Acceptance**

Reviewer #1: No more concerns

Reviewer #2: None

**Part III – Minor Issues: Editorial and Data Presentation Modifications**

Reviewer #1: no

Reviewer #2: None. The authors addressed my previous concerns and submitted an improved manuscript.

PLOS authors have the option to publish the peer review history of their article (what does this mean?). If published, this will include your full peer review and any attached files.

Reviewer #1: **Yes: **Estaquier Jérôme

Reviewer #2: No

---

## [Editor Report · Acceptance letter]

4 May 2021

Dear Dr. Sodora,

We are delighted to inform you that your manuscript, "Rapid progression is associated with lymphoid follicle dysfunction in SIV-infected infant rhesus macaques.," has been formally accepted for publication in PLOS Pathogens.

Best regards,

Kasturi Haldar

Editor-in-Chief

PLOS Pathogens

orcid.org/0000-0001-5065-158X

Michael Malim

Editor-in-Chief

PLOS Pathogens

orcid.org/0000-0002-7699-2064